# The Link of Pancreatic Iron with Glucose Metabolism and Cardiac Iron in Thalassemia Intermedia: A Large, Multicenter Observational Study

**DOI:** 10.3390/jcm10235561

**Published:** 2021-11-26

**Authors:** Antonella Meloni, Laura Pistoia, Maria Rita Gamberini, Paolo Ricchi, Valerio Cecinati, Francesco Sorrentino, Liana Cuccia, Massimo Allò, Riccardo Righi, Priscilla Fina, Ada Riva, Stefania Renne, Giuseppe Peritore, Stefano Dalmiani, Vincenzo Positano, Emilio Quaia, Filippo Cademartiri, Alessia Pepe

**Affiliations:** 1Department of Radiology, Fondazione Toscana Gabriele Monasterio, 56124 Pisa, Italy; antonella.meloni@ftgm.it (A.M.); miot@ftgm.it (L.P.); positano@ftgm.it (V.P.); fcademartiri@ftgm.it (F.C.); 2Unità Operativa Complessa di Bioingeneria, Fondazione Toscana Gabriele Monasterio, 56124 Pisa, Italy; 3Unità Operativa di Day Hospital della Talassemia e delle Emoglobinopatie, Dipartimento della Riproduzione e dell’Accrescimento, Azienda Ospedaliero-Universitaria “S. Anna”, 44124 Cona, Italy; m.gamberini@ospfe.it; 4Unità Operativa Semplice Dipartimentale Malattie Rare del Globulo Rosso, Azienda Ospedaliera di Rilievo Nazionale, 80131 Napoli, Italy; paolo.ricchi@aocardarelli.it; 5Struttura Semplice di Microcitemia, Ospedale “SS. Annunziata” Azienda Sanitaria Locale Taranto, 74123 Taranto, Italy; valerio.cecinati@asl.taranto.it; 6Unità Operativa Semplice Dipartimentale Day Hospital Talassemici, Ospedale “Sant’Eugenio”, 00143 Roma, Italy; francescosorrentino1@alice.it; 7Unità Operativa Complessa Ematologia con Talassemia, Azienda di Rilievo Nazionale ad Alta Specializzazione Civico “Benfratelli-Di Cristina”, 90127 Palermo, Italy; liana.cuccia@arnascivico.it; 8Ematologia Microcitemia, Ospedale San Giovanni di Dio—Azienda Sanitaria Provinciale di Crotone, 88900 Crotone, Italy; allomassimo@libero.it; 9Diagnostica per Immagini e Radiologia Interventistica, Ospedale del Delta, 44023 Lagosanto, Italy; riccardo.righi@ausl.fe.it; 10Unità Operativa Complessa Diagnostica per Immagini, Ospedale “Sandro Pertini”, 00157 Roma, Italy; priscilla.fina@gmail.com; 11Struttura Complessa di Radiologia, Ospedale “SS. Annunziata” Azienda Sanitaria Locale Taranto, 74100 Taranto, Italy; ada.riva@yahoo.it; 12Struttura Complessa di Cardioradiologia-Unità di Terapia Intensiva Cardiologica, Presidio Ospedaliero “Giovanni Paolo II”, 88046 Lamezia Terme, Italy; stefania.renne@virgilio.it; 13Unità Operativa Complessa di Radiologia, Azienda di Rilievo Nazionale ad Alta Specializzazione Civico “Benfratelli-Di Cristina”, 90127 Palermo, Italy; giuseppe.peritore@hotmail.it; 14Unità Operativa Complessa Infotelematica Translational BioInformatics and eHealth, Fondazione Toscana Gabriele Monasterio, 56124 Pisa, Italy; dalmiani@ftgm.it; 15Department of Medicine, Institute of Radiology, University of Padua, 35100 Padua, Italy; emilio.quaia@unipd.it

**Keywords:** thalassemia intermedia, pancreas, iron overload, magnetic resonance imagining, glucose metabolism

## Abstract

In thalassemia major, pancreatic iron was demonstrated as a powerful predictor not only for the alterations of glucose metabolism but also for cardiac iron, fibrosis, and complications, supporting a profound link between pancreatic iron and heart disease. We determined for the first time the prevalence of pancreatic iron overload (IO) in thalassemia intermedia (TI) and systematically explored the link between pancreas T2* values and glucose metabolism and cardiac outcomes. We considered 221 beta-TI patients (53.2% females, 42.95 ± 13.74 years) consecutively enrolled in the Extension–Myocardial Iron Overload in Thalassemia project. Magnetic Resonance Imaging was used to quantify IO (T2* technique) and biventricular function and to detect replacement myocardial fibrosis. The glucose metabolism was assessed by the oral glucose tolerance test (OGTT). Pancreatic IO was more frequent in regularly transfused (*N* = 145) than in nontransfused patients (67.6% vs. 31.6%; *p* < 0.0001). In the regular transfused group, splenectomy and hepatitis C virus infection were both associated with high pancreatic siderosis. Patients with normal glucose metabolism showed significantly higher global pancreas T2* values than patients with altered OGTT. A pancreas T2* < 17.9 ms predicted an abnormal OGTT. A normal pancreas T2* value showed a 100% negative predictive value for cardiac iron. Pancreas T2* values were not associated to biventricular function, replacement myocardial fibrosis, or cardiac complications. Our findings suggest that in the presence of pancreatic IO, it would be prudent to initiate or intensify iron chelation therapy to prospectively prevent both disturbances of glucose metabolism and cardiac iron accumulation.

## 1. Introduction

Beta-thalassemias are one of the most common monogenic disorders worldwide, caused by reduced or absent synthesis of the beta chains of hemoglobin [1]. Based on the clinical presentation, three main β-thalassemia phenotypes are conventionally recognized: major, intermedia, and minor. β-thalassemia major (TM) represents the most severe form, characterized by early presentation (before the 2 years of age) of symptoms and the need for regular lifelong blood transfusions for survival [2]. Beta-thalassemia minor, also called carrier or trait, is associated with the inheritance of a single gene defect and is usually asymptomatic, with mild to moderate microcytic anemia. β-thalassemia intermedia (TI) entails a broad spectrum of clinical presentations. Some patients remain asymptomatic for most of their lives, with hemoglobin levels ranging between 7 and 10 g/dL, and require occasional transfusions in specific circumstances such as pregnancy, surgery, or acute infection, while other patients are committed to receive regular blood transfusions for normal sustained growth, for the prevention or management of complications, or to compensate the age-related changes in adaptation to anemia and the difficulty in maintaining a high output with normal vascular aging [3,4,5,6].

The triad of ineffective erythropoiesis, chronic anemia, and iron overload (IO) is at the core of all TI-related morbidities. In β-TI, iron overload is primarily due to increased intestinal iron absorption and iron release from macrophages triggered by inappropriately low levels of serum hepcidin [7], but also chronic hemolysis and blood transfusions can play a role [8]. Although occurring at a much slower rate than in TM patients, iron overload, with its cytotoxic effects, represents an important concern also in β-TI. Indeed, patients with TI can develop a variety of IO-related hepatic, cardiac, and endocrine complications [5]. So, a careful and noninvasive monitoring of organ-specific IO can be an important further step toward a better management of TI patients. The T2* Magnetic Resonance Imaging (MRI) technique perfectly fits to this purpose. The literature on T2* MRI in TI patients is scarce, especially in comparison with the large number of studies available in TM, but it has been demonstrated that while hepatic iron accumulation is a frequent issue, cardiac siderosis is uncommon [9,10,11]. Moreover, it has been shown that elevated liver iron concentration (LIC) by MRI is a marker of increased vascular, endocrine, and bone disease in β-TI patients [12].

To the best of our knowledge, no study has evaluated in TI the prevalence of pancreatic IO and its clinical correlates. The impaired insulin excretory function secondary to chronic iron overload in the pancreas is one of the main determinants of altered glucose metabolism in thalassemia [13]. In β-TI patients, glucose intolerance occurs early during adolescence and in a not recent study its prevalence was demonstrated to be about 24% [14]. Diabetes frequently occurs at later stages and in nontransfused patients older than 32 years its incidence can reach 13% [3]. So, it is important to understand if pancreatic T2* measurements can have a role in identifying the patients at high risk for glucose dysregulation in this population. In TM, pancreatic iron was demonstrated to be the strongest predictor of beta cell toxicity [15] and, in a large cohort of well-transfused and chelated patients, a normal global pancreas T2* value showed a negative predictive value of 100% for disturbances of glucose metabolism [16]. Moreover, in TM, pancreatic iron was demonstrated as a powerful predictor not only for the alterations of glucose metabolism but also for cardiac iron and complications, supporting a profound link between pancreatic iron and heart disease [16,17]. However, these findings in TM patients cannot be directly translated in TI, due to the dissimilar molecular background and the different treatment approach (splenectomy, transfusion, iron chelation). Conversely to TM, where myocardial iron overload plays a significant role, in TI the main causative factors of heart disease are chronic tissue hypoxia with a consequent high cardiac output state [18].

Based on this background, the goals of this multicenter study were to determine the prevalence of pancreatic IO in β-TI and to systematically explore in this population—stratified based on the transition to a regular transfusion regimen—the link between pancreas T2* values and glucose metabolism and cardiac iron, function, and complications.

## 2. Materials and Methods

### 2.1. Study Population

We considered 221 β-TI patients (53.2% females, mean age: 42.95 ± 13.74 years, age range: 9–77 years), consecutively enrolled in the Extension–Myocardial Iron Overload in Thalassemia (E-MIOT) project between October 2015 and March 2020. E-MIOT is an Italian network constituted by 66 thalassemia centers and 11 MRI sites where MRI exams are performed using homogeneous, standardized, and validated procedures for the heart, the liver, and the pancreas [19,20,21]. All centers use a web-based central system for collection of clinical, laboratory, and instrumental data, implementing the applicable requirements from General Data Protection Regulation (European Union 2016/679) and Directive on security of network and information systems (European Union 2016/1148).

The study complied with the Declaration of Helsinki and was approved by the ethical committees of all the MRI sites involved in the study. All patients gave written informed consent.

### 2.2. MRI Protocol

MRI exams were performed on conventional clinical 1.5T scanners of three main vendors (GE Healthcare, Milwaukee, WI, United States of America; Philips Healthcare, Best, The Netherlands; Siemens Healthineers; Erlangen, Germany), equipped with a phased-array receiver surface coil.

For iron overload assessment, five or more axial slices including the whole pancreas [22], a mid-transverse hepatic slice [23], and basal, medium, and apical short-axis views of the left ventricle (LV) [24,25] were acquired by T2* gradient-echo multiecho sequences. T2* image analysis was performed using a custom-written, previously validated software (HIPPO MIOT^®^, Version 2.0, Consiglio Nazionale delle Ricerche and Fondazione Toscana Gabriele Monasterio, Pisa, Italy, 2015) [26]. Three small regions of interest (ROIs) were manually defined over pancreatic head, body, and tail, encompassing parenchymal tissue and taking care to avoid large blood vessels or ducts and areas involved in susceptibility artefacts from gastric or colic intraluminal gas [27]. Global pancreatic T2* value was calculated as the mean of T2* values from the three regions. Hepatic T2* values were calculated in a circular ROI of standard dimension [28] and were converted into liver iron concentration (LIC) using Wood’s calibration curve [29,30]. The software provided the T2* value on each of the 16 segments of the LV, according to the standard American Heart Association/American College of Cardiology model [31]. The global heart T2* value was obtained by averaging all segmental T2* values. The reproducibility of the methodology had been previously assessed for all organs [21,32].

Steady-state free precession (SSFP) cine images were acquired in sequential 8-mm short-axis slices from the atrio-ventricular ring to the apex to quantify biventricular function parameters in a standard way [33]. The intercenter variability had been previously reported [34].

To detect the presence of replacement myocardial fibrosis, late gadolinium enhancement (LGE) short-axis and vertical, horizontal, and oblique long-axis images were acquired 10–18 min after Gadobutrol (Gadovist^®^; Bayer; Berlin, Germany) intravenous administration at the standard dose of 0.2 mmol/kg using a fast gradient–echo inversion recovery sequence. The use of Gadobutrol has been demonstrated to be safe in patients with hemoglobinopathies [35]. LGE images were not acquired in patients with a glomerular filtration rate <30 mL/min/1.73 m^2^ and in patients who refused the contrast medium administration. LGE was considered present when visualized in two different views [36].

### 2.3. Assessment of Glucose Metabolism

To assess the disturbances of glucose metabolism, patients not already diagnosed with diabetes performed an oral glucose tolerance test (OGTT) within three months from the MRI study at the reference thalassemia center.

All patients were required to fast overnight (at least 12 h). Baseline blood assessments of glucose and insulin were performed. Patients were given 1.75 g/kg (maximum dose of 75 g) of glucose solution and glucose and insulin were measured at 60 and 120 min. In the patients without known diabetes, we used the homeostasis model assessment (HOMA) of insulin resistance (HOMA-IR) index to assess the insulin resistance and it was calculated as the product of fasting glucose and insulin levels divided by 405 [37]. The HOMA of β-cell function (HOMA-B) index was used as measure of β-cell function and computed as the product of 360 and fasting insulin levels divided by the value of fasting glucose concentrations minus 63 [38].

### 2.4. Diagnostic Criteria

Twenty-six ms was previously demonstrated to be the lowest threshold of normal global pancreas T2* value [22]. A LIC <3 mg/g/dw indicated no significant hepatic IO [39]. The value of 20 ms was used as “conservative” normal value for segmental and global heart T2* values [40].

A fasting plasma glucose of less than 100 mg/dL and 2-h glucose of less than 140 mg/dL indicated normal glucose tolerance (NGT). Impaired fasting glucose (IFG) was diagnosed in presence of fasting plasma glucose levels between 100 and 126 mg/dL. Impaired glucose tolerance (IGT) was defined by 2-h plasma glucose between 140–200 mg/dL, with a fasting plasma glucose <126 mg/dL. Diabetes mellitus (DM) was defined by fasting plasma glucose ≥126 mg/dL or 2-h plasma glucose ≥200 mg/dL during an OGTT or a random plasma glucose ≥200 mg/dL with classic symptoms of hyperglycemia or hyperglycemic crisis [41].

Heart failure was diagnosed by clinicians based on symptoms, signs, and instrumental findings according to the current guidelines [42]. Arrhythmias were diagnosed only if documented by electrocardiogram and requiring specific medication. Arrhythmias were classified according to the American Heart Association/American College of Cardiology Guidelines [43]. Pulmonary hypertension was diagnosed if the trans-tricuspidal velocity jet was greater than 3.2 m/s [44]. The term “cardiac complications” included heart failure, arrhythmias, and pulmonary hypertension clinically active at the time of the MRI.

### 2.5. Statistical Analysis

All data were analyzed using SPSS version 27.0 (IBM Corp., Armonk, NY, USA) statistical package.

Continuous variables were described as mean ± standard deviation (SD) and categorical variables were expressed as frequencies and percentages.

For continuous values with normal distribution, comparisons between groups were made by independent-samples t-test (for 2 groups) or one-way analysis of variance-ANOVA (for more than 2 groups). Wilcoxon’s signed rank test or Kruskal–Wallis test were applied for continuous values with no normal distribution. The χ^2^ test was used for the comparison of noncontinuous variables. Bonferroni post hoc test was used for multiple comparisons between pairs of groups.

Correlation analysis was performed using Pearson’s test or Spearman’s test where appropriate.

To determine the best glucose and pancreas T2* cut-offs for discriminating the presence of an altered OGTT, the maximum sum of sensitivity and specificity was calculated from receiver operating characteristic (ROC) curve analysis.

In all tests, a 2-tailed probability value of 0.05 was considered statistically significant.

## 3. Results

### 3.1. Patients’ Characteristics

One-hundred and forty-five (65.6%) patients were under regular transfusion therapy (>4 transfusions per year), starting at a mean age of 15.29 ± 17.62 years.

Table 1 shows the comparison between nontransfused and regularly transfused TI patients (NT-TI and RT-TI, respectively).

Age and sex were comparable between the two groups while RT-TI patients were more frequently splenectomized and chelated and showed a higher frequency of past/active HCV infection and significantly higher serum hemoglobin and ferritin values. Besides the lower frequency of hepatic iron overload, RT-TI patients had more pancreatic iron overload (Figure 1). Left ventricular end-diastolic volume index and mass index were significantly lower in RT-TI patients.

### 3.2. Pancreatic Iron and Demographics

The youngest patient with pancreatic IO was a female of 11 years old. She was transfused since 6 years of age and did not have hepatic IO.

Frequency of pancreatic IO was comparable between males and females in NT-TI patients (34.1% vs. 28.6%; *p* = 0.602) as well as in RT-TI patients (62.9% vs. 71.1%; *p* = 0.298). No correlation was detected between global pancreas T2* values and age (NT-TI: R = −0.156, *p* = 0.178 and RT-TI: R = 0.012, *p* = 0.891).

In the group of NT-TI patients global pancreas T2* values were comparable between patients without and with the spleen (27.94 ± 10.32 ms vs. 30.11 ± 9.44 ms; *p* = 0.342), while among the RT-TI patients significantly lower global pancreas T2* were detected in the splenectomized group (18.59 ± 11.29 ms vs. 24.16 ± 13.39; *p* = 0.024).

Independently from the transfusional state, no correlation was detected between global pancreas T2* values and mean serum ferritin levels (NT-TI: R = 0.026, *p* = 0.830 and RT-TI: R = −0.140, *p* = 0.128).

Global pancreas T2* values were lower in NT-TI patients with an active or eradicated HCV infection than in negative NT-TI patients, but statistical significance was not reached (23.72 ± 9.29 ms vs. 29.65 ± 9.83 ms; *p* = 0.124), while among the RT-TI patients, the presence of an active or eradicated HCV infection was associated with significantly lower global pancreas T2* (16.22 ± 10.40 ms vs. 22.34 ± 12.45 ms; *p* = 0.005).

### 3.3. Pancreatic Iron and Glucose Metabolism

Disturbances of glucose metabolism were found in 12 (15.8%) NT-TI patients (2 IFG, 8 IGT, and 2 DM) and in 33 (22.8%) RT-TI patients (8 IFG, 18 IGT, and 7 DM). The prevalence of DM in the whole patient cohort was 4.1%.

In NT-TI patients, a fasting plasma glucose >84 mg/dL predicted the presence of an abnormal OGTT with a sensitivity of 72.7% and a specificity of 66.1% (*p* = 0.050) and the area under the curve was 0.71 (95% confidence interval-CI: 0.59–0.82). In RT-TI patients, a fasting plasma glucose >91 mg/dL predicted the presence of an abnormal OGTT with a sensitivity of 76.7% and a specificity of 79.3% (*p* < 0.0001) and the area under the curve was 0.82 (95% CI: 0.74–0.87).

Table 2 shows the association of pancreatic T2* values and MRI LIC values with glucose and insulin levels evaluated during the OGTT in patients without DM. In RT-TI patients, pancreatic T2* values were inversely correlated with fasting and 1-hr glucose levels and positively associated with the HOMA of β-cell function index. No association was found between pancreatic or hepatic iron and HOMA-IR index.

The 34.4% of NT-TI patients and the 63.4% of RT-TI patients had a normal glucose metabolism, but a pathological global pancreas T2* value.

In the group of NT-TI patients, global pancreas T2* values were comparable between patients with normal and altered glucose metabolism (29.85 ± 11.41 ms vs. 28.87 ± 9.67 ms; *p* = 0.755). The ROC curve did not reveal a global pancreas T2* threshold below which the probability of detecting an altered glucose metabolism increases significantly with satisfying sensitivity and specificity (AUC = 0.52; *p* = 0.823).

In the group of RT-TI patients, the presence of disturbances of glucose metabolism was associated with significantly lower global pancreas T2* (Figure 2A) and from ROC curve analysis, a global pancreas T2* < 17.9 ms predicted the presence of an abnormal OGTT with a sensitivity of 69.7% and a specificity of 56.2% (*p* = 0.045). The area under the curve was 0.61 (95% CI: 0.53–0.69) (Figure 2B).

### 3.4. Pancreas T2* and MRI Correlates

Globally, out of the 122 patients with pancreatic iron overload, 66 (54.1%) had a normal MRI LIC value. A significant inverse correlation between MRI LIC and pancreatic T2* values was found only in RT-TI patients (R = −0.244, *p* = 0.003).

Global pancreas T2* values showed a significant positive correlation with global heart T2* values in RT-TI patients (R = 0.193, *p* = 0.020) (Figure 3A) and a significant negative correlation with the number of segments with abnormal T2* in both NT-TI patients (R = −0.264, *p* = 0.001) and RT-TI patients (R = −0.268, *p* = 0.001). Pancreatic iron overload was detected in all 5 patients (1 NT and 4 RT) with significant myocardial IO. A normal global pancreas T2* value showed a negative predictive value of 100% for cardiac iron. Based on the segmental approach, 199 (90.0%) TI patients showed no myocardial IO (all segments with T2* ≥ 20 ms), 17 (7.7%) showed a heterogeneous iron distribution (some segments with T2* ≥ 20 ms and others with T2* < 20 ms) with global heart T2* ≥ 20 ms, 5 (2.3%) showed a heterogeneous myocardial IO with global heart T2* < 20 ms, and none showed a homogeneous myocardial IO (all segments with T2* < 20 ms). The distribution of the patterns of myocardial IO was comparable between NT-TI and RT-TI patients (*p* = 0.790). Patients with no myocardial IO showed significantly higher global pancreas T2* values than patients with a heterogeneous myocardial IO with global heart T2* < 20 ms (23.91 ± 11.81 ms vs. 8.17 ± 8.61 ms; *p* = 0.012) (Figure 3B).

LV and right ventricular (RV) end-diastolic volume indexes and ejection fractions and LV mass index were not correlated with global pancreatic T2* values.

Macroscopic myocardial fibrosis was detected in 26.5% of the 82 patients in which the contrast medium was administered. Global pancreas T2* values were comparable in patients without and with myocardial fibrosis in NT-TI patients (27.38 ± 9.95 ms vs. 27.82 ± 10.68 ms; *p* = 0.530) as well as in RT-TI patients (22.54 ± 11.57 ms vs. 14.54 ± 14.03 ms; *p* = 0.054), although in the latter group, the *p* -value was close to statistical significance.

### 3.5. Pancreatic Iron and Cardiac Complications

Five (6.6%) NT-TI patients had at least one cardiac complication: 2 heart failure, 1 supraventricular arrhythmia, 1 supraventricular arrhythmias + pulmonary hypertension, and 1 pulmonary hypertension. Out of the two patients with heart failure, one had pancreatic IO while none had significant myocardial IO (global heart T2* < 20 ms).

Eighteen (12.4%) RT-TI patients had at least one cardiac complication: 1 heart failure, 11 arrhythmias (9 supraventricular and 2 ventricular), 2 supraventricular arrhythmias + pulmonary hypertension, and 8 pulmonary hypertension. RT-TI patients without and with cardiac complications showed comparable global heart T2* values (40.43 ± 4.24 ms vs. 39.22 ± 6.55 ms; *p* = 0.320) and global pancreas T2* values (21.02 ± 13.97 ms vs. 19.78 ± 11.78 ms; *p* = 0.310).

## 4. Discussion

To the best of our knowledge, no studies in the literature have previously quantified pancreatic iron by T2* technique in β-TI. In the current study, pancreatic iron overload was detected in almost one third of never or sporadically transfused TI patients and its prevalence was more than double in RT-TI patients, confirming that the prevalence of pancreatic IO increases with regular transfusions. The apparently controversial finding of a higher frequency of hepatic IO among NT-TI vs. RT-TI patients could be explained by lower frequency of chelation therapy in the first group. Iron is removed from different organs at different rates [45] and, while hepatic iron levels can be removed in few months, it seems extremely hard to remove iron from the pancreas. As expected, pancreatic iron loading in transfused TI patients occurs later in life and to a lesser degree than for TM patients [16], principally due to the lower duration and intensity of transfusion therapy.

In agreement with studies focused on TM patients [16,46], we found a higher pancreatic siderosis in splenectomized vs. nonsplenectomized RT-TI patients. The spleen acts as a store for nontoxic iron, protecting the rest of the body from this iron [47,48].

Besides the lower prevalence of HCV infection in comparison with TM patients [16], the association between HCV infection and higher levels of pancreatic iron deposition was confirmed also for TI patients. Different from the RT-TI patients, in the subgroup of NT-TI patients, the statistical significance was not reached probably due to the low number of positive cases. This link is likely caused by the capacity of the HCV to induce a decrease in hepcidin levels [49]. In addition, it has been demonstrated that hepcidin can be expressed also by pancreatic β-cells [50] and that HCV can be present in human pancreatic β-cells [51].

The prevalence of disturbances of glucose metabolism tended to be higher in RT-TI than in NT-TI patients but without statistical significance and, in these two groups, a fasting plasma glucose >91 mg/dL and >84 mg/dL, respectively, identified an abnormal OGTT. These cut-offs are lower to those previously identified in TM, corresponding to 97–98 mg/dL [15,16], and seem to suggest that lower cut-offs should be used to increase specificity of the diagnosis in TI. The higher overall prevalence of DM in our study population compared with that reported in the OPTIMAL CARE -Overview on Practices in Thalassemia Intermedia Management Aiming for Lowering Complication rates Across a Region of Endemicity- study (4.1% vs. 1.7%) may be explained by the higher mean age (42.99 ± 13.72 years vs. 25.44 ± 13.86 years) [4]. In nondiabetic NT-TI patients, pancreatic T2* values were not correlated to glucose levels, likely because most of the patients had normal or only slightly altered values of these parameters. Conversely, in RT-TI patients, a weak inverse correlation between the HOMA-B—reflecting the fasting β-cell function—and the increased iron deposition in the pancreas was detected. The lack of an association between insulin resistance index and pancreatic T2* values mirrors previous observations in TM and is not fully unanticipated [16]. In fact, the use of indices based on fasting glucose and insulin concentrations may mask the truly increased insulin resistance. Moreover, in RT-TI, as in TM patients, the glucose homeostasis is a complex phenomenon where both impaired insulin release, caused by iron deposition in the pancreatic cells [13], and insulin resistance, caused by hepatic iron deposition interfering with insulin’s ability to suppress hepatic glucose production and iron deposition in the muscle decreasing glucose uptake, play a role [52]. In the nontransfused group, the small number of patients with an altered glucose metabolism probably hid its association with reduced pancreatic T2* values. In the regularly transfused group, patients with normal glucose metabolism showed significantly higher global pancreas T2* values than patients with disturbances of glucose metabolism, and a pancreatic T2* < 17.9 ms emerged as the best cut-off for predicting an abnormal glucose metabolism. The AUC was rather low and the sensitivity was moderate, since genetic, behavioral, and environmental factors can contribute to glucose dysregulation [53] and also hypoxia can promote the insulin resistance [54]. The moderate specificity can be explained by the fact that a latency time exists before pancreatic iron could provide IGT and overt DM. Anyway, this cut-off may help to select the patients in which an intensification or modification of the iron chelation therapy may prospectively prevent the advent of clinical disease.

Our study showed that in the clinical practice, pancreatic IO cannot be estimated by serum ferritin levels. Moreover, hepatic and pancreatic T2* values were not correlated in NT-TI patients and only weakly correlated in RT-TI patients, confirming the importance to quantify iron status in the different organs by MRI.

We detected a significant correlation between hemosiderosis of the pancreas and heart, which is more likely due to the same L-type calcium iron channels in the two organs, taking up circulating non-transferrin-bound iron [55]. A normal global pancreas T2* value showed a negative predictive value of 100% for cardiac iron. So, not only in TM where cardiac iron is a major issue [16,17,56] but also in TI, pancreatic T2* measurements could serve as an early warning system for cardiac iron loading and should be routinely obtained.

Despite the link between pancreatic and cardiac iron, we failed to detect a correlation between pancreatic T2* values and cardiac function, likely because in TI the cardiac involvement is mainly related to the long-term exposure to chronic anemia and tissue hypoxia determining a high cardiac output state cardiomyopathy [18].

In TM patients, pancreatic iron burden was demonstrated correlated with replacement myocardial fibrosis and cardiac complications [16]. Indeed, HCV infection, associated with higher levels of pancreatic IO, is one of the possible mechanisms for replacement myocardial fibrosis through both myocarditis directly [36,57] and the pancreas and liver damage with the development of DM indirectly [58,59]. Moreover, replacement myocardial fibrosis emerged as the strongest MRI predictor for heart failure and cardiac complications in TM [60]. In our TI population, we found lower pancreas T2* among positive-LGE patients but, likely due to the lower number of cases, statistical significance was not reached. We failed to detect an association between pancreatic iron burden and the presence of cardiac complications, likely because the most frequently detected complications were supraventricular arrhythmias and pulmonary hypertension, less-associated to iron overload than heart failure. In addition, since significantly lower T2* values were found in TM [16], we may speculate that a more prolonged and severe pancreatic iron exposition is needed to determine the development of cardiac complications.

### Limitations

One of the main limitations of our study was the low number of NT-TI patients.

Although the HOMA-B index reflects fasting β-cell function, it cannot provide insight into the secretory response of beta cells to rising glucose concentrations.

Longitudinal prospective studies involving more patients are needed to better clarify the temporal association between pancreatic iron and pancreatic dysfunction and cardiac iron.

## 5. Conclusions

NT-TI patients have lower pancreatic iron loading than RT-TI patients, besides the higher liver iron level due to the less frequent exposure to iron chelation therapy. In RT-TI patients, pancreatic iron is a predictor for the alterations of glucose metabolism and for cardiac iron. If a patient has pancreatic iron overload, it would be prudent to initiate or intensify iron chelation therapy to prospectively prevent both alterations of glucose metabolism and cardiac iron accumulation, thus, reducing the risk for full-blown DM and cardiac complications.

## Figures and Tables

**Figure 1 jcm-10-05561-f001:**
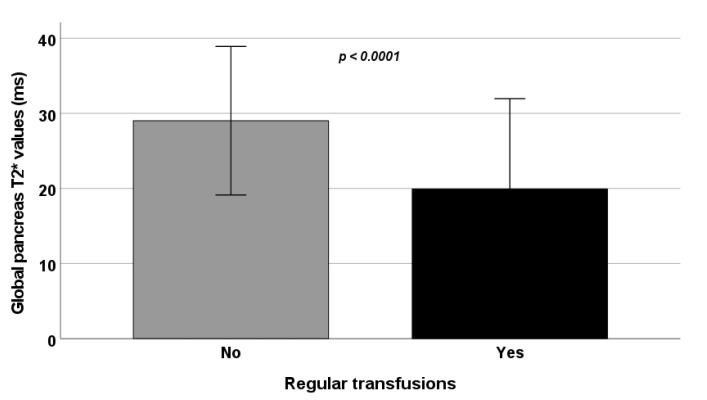
Comparison of global pancreas T2* values between NT-TI and RT-TI patients.

**Figure 2 jcm-10-05561-f002:**
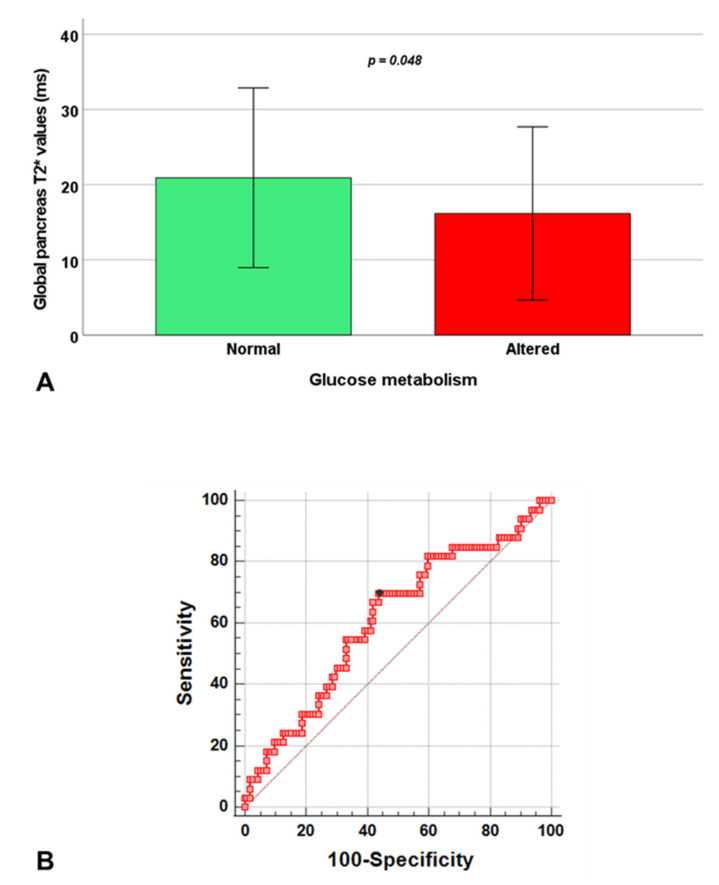
Association between global pancreas T2* values and glucose metabolism in RT-TI patients. (**A**) Global pancreas T2* values in patients with normal and altered glucose metabolism. (**B**) ROC curve analysis of global pancreas T2* values to predict an abnormal oral glucose tolerance test.

**Figure 3 jcm-10-05561-f003:**
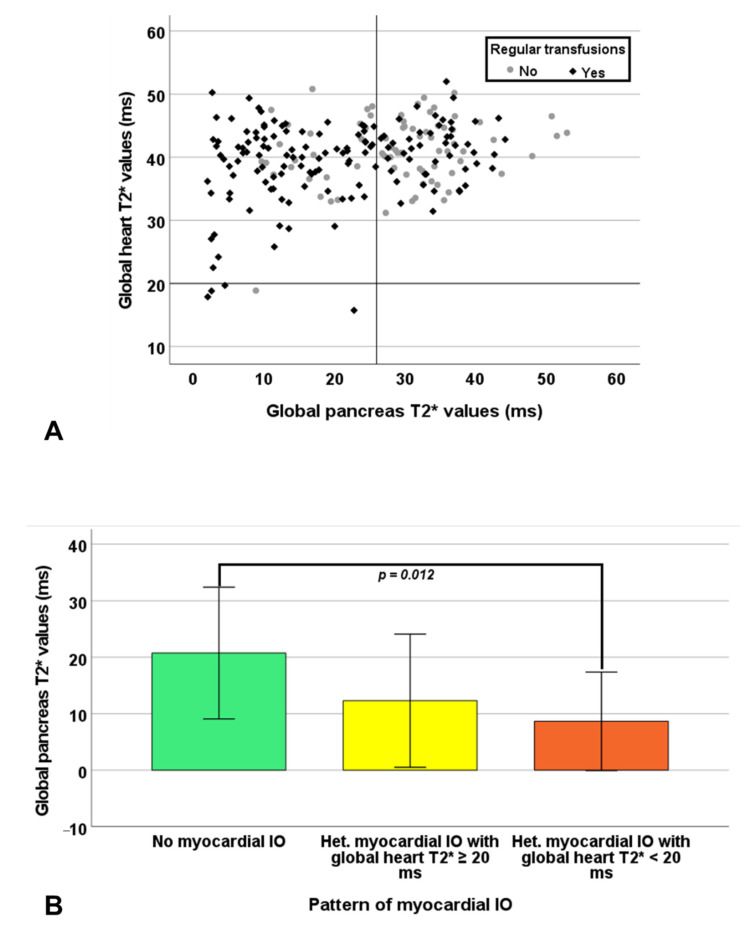
(**A**) Scatter plot of global heart T2* values versus global pancreas T2*values. The horizontal and vertical dotted lines represent the cut-off for T2* values. (**B**) Global pancreas T2* values in the groups of patients with different patterns of myocardial iron overload.

**Table 1 jcm-10-05561-t001:** Comparison of demographic, clinical, and MRI data between nontransfused and regularly transfused and thalassemia intermedia patients.

	NT-TI Patients (*N* = 76)	RT-TI Patients (*N* = 145)	*p*-Value
Age (years)	43.99 ± 13.87	42.47 ± 13.66	0.435
Females, N (%)	35 (46.1)	83 (57.2)	0.113
Splenectomy, N (%)	38 (50.0)	110 (75.9)	<0.0001
HCV infection, N (%)	8 (10.5)	57 (39.3)	<0.0001
Serum hemoglobin (g/dL)	9.15 ± 1.09	9.48 ± 0.57	0.007
Mean serum ferritin (ng/mL)	515.87 ± 472.29	827.01 ± 876.45	0.001
Chelated, N (%)	37 (48.7)	139 (95.9)	<0.0001
Altered glucose metabolism, N (%)	12 (15.8)	33 (22.8)	0.222
MRI LIC (mg/g dL)	5.88 ± 6.28	6.02 ± 13.86	0.091
Hepatic IO, N (%)	44 (57.9)	56 (38.6)	0.006
Global heart T2* (ms)	41.13 ± 5.29	39.37 ± 6.31	0.076
Myocardial IO, N (%)	1 (1.3)	4 (2.8)	0.662
Global pancreas T2* (ms)	29.02 ± 9.89	19.93 ± 12.03	<0.0001
Pancreatic IO, N (%)	24 (31.6)	98 (67.6)	<0.0001
LV end-diastolic volume index (mL/m^2^)	91.70 ± 18.86	86.07 ± 15.49	0.029
LV mass index (g/m^2^)	60.92 ± 15.11	56.69 ± 12.91	0.016
LV ejection fraction (%)	63.09 ± 6.93	63.37 ± 7.27	0.786
RV end-diastolic volume index (mL/m^2^)	86.28 ± 19.18	82.78 ± 16.24	0.306
RV ejection fraction (%)	63.58 ± 7.15	63.74 ± 7.89	0.926
Replacement myocardial fibrosis, N (%)	11/35 (31.4)	11/45 (24.4)	0.488

NT, nontransfused; TI, thalassemia intermedia; N, number; RT, regularly transfused; HCV, hepatitis C virus; MRI, magnetic resonance imaging; LIC, liver iron concentration; IO, iron overload; LV, left ventricular; RV, right ventricular.

**Table 2 jcm-10-05561-t002:** Association of pancreas T2* values and MRI LIC values with glucose and insulin concentrations.

Variable	Mean Value	Correlation with Global Pancreas T2* Values		Correlation with MRI LIC Values	
		R	*p*-Value	R	*p*-Value
Nontransfused TI patients without diabetes					
Fasting plasma glucose (mg/dL)	79.39 ± 11.80	−0.019	0.879	0.195	0.119
1-hr plasma glucose after OGTT (mg/dL)	135.71 ± 32.17	−0.061	0.693	0.184	0.227
2-hr plasma glucose after OGTT (mg/dL)	108.69 ± 25.09	−0.142	0.337	−0.079	0.593
Fasting plasma insulin (µU/mL)	6.65 ± 11.04	−0.013	0.927	−0.078	0.579
1-hr plasma insulin after OGTT (µU/mL)	29.97 ± 19.07	0.224	0.404	−0.063	0.816
2-hr plasma insulin after OGTT (µU/mL)	19.57 ± 19.79	0.474	0.064	−0.337	0.202
HOMA-IR index	1.29 ± 1.85	0.167	0.237	−0.083	0.560
HOMA-B index (%)	133.66 ± 132.82	0.150	0.308	−0.159	0.281
Regularly transfused TI patients without diabetes					
Fasting plasma glucose (mg/dL)	84.86 ± 12.32	−0.335	<0.0001	0.162	0.088
1-hr plasma glucose after OGTT (mg/dL)	135.71 ± 32.17	−0.308	0.016	0.156	0.229
2-hr plasma glucose after OGTT (mg/dL)	108.69 ± 25.09	−0.129	0.097	0.098	0.429
Fasting plasma insulin (µU/mL)	8.14 ± 10.29	0.204	0.073	0.005	0.964
1-hr plasma insulin after OGTT (µU/mL)	47.91 ± 24.31	−0.243	0.383	−0.193	0.491
2-hr plasma insulin after OGTT (µU/mL)	32.15 ± 20.12	−0.068	0.810	0.068	0.810
HOMA-IR index	1.67 ± 2.09	0.125	0.277	0.042	0.716
HOMA-B index (%)	277.13 ± 486.21	0.391	0.001	−0.141	0.233

MRI, magnetic resonance imaging; LIC, liver iron concentration; TI, thalassemia intermedia; HOMA-IR, homeostasis model assessment for insulin resistance; HOMA-B, homeostasis model assessment for β-cell function.

## Data Availability

The data presented in this study are available on request from the corresponding author. The data are not publicly available due to privacy.

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
