# Peer review of "The Link of Pancreatic Iron with Glucose Metabolism and Cardiac Iron in Thalassemia Intermedia: A Large, Multicenter Observational Study"

_jcm, 2021, doi:10.3390/jcm10235561_

Round 1

Reviewer 1 Report

In the manuscript entitled "The link of pancreatic iron with glucose metabolism and cardiac iron in thalassemia intermedia: a large, multicenter observational study" Meloni et al. aim to determine the link between pancreatic iron and glucidic metabolism, cardiac iron and complications through T2* MRI values. The study is well designed and provides novel insights regarding the use of pancreatic iron overload for clinical use in the context of thalassemia intermedia. The manuscript is generally well written, but this Reviewer has some minor remarks that should be considered before publication.

  1. Since the aim of this study and its findings are quite novel, the abstract could be strengthened by the use of a small piece of “introductory” sentences (one or two sentences).
  2. Although all abbreviations have been determined this Reviewer believes that in some sentences there is an extensive use of these abbreviations, making the manuscript more difficult to read. I believe that using a slightly smaller number of abbreviations would help the reader follow the manuscript.
  3. In Figure 3 please provide a title.
  4. Some AUCs are rather low to moderate (e.g., 0.71 and 0.61). In the second case, the authors have to be praised for admitting this low to moderate cutoff in the Discussion section. However, they have to point it out in the first case for the group of NT-TI patients as well.
  5. This Reviewer believes that some parentheses containing numbers in the Discussion section could be avoided to make the manuscript easier to follow.

Author Response

We would like to thank the Editor and the Reviewer for their encouraging feedback and constructive critique and for the effort regarding this manuscript. We have addressed each of the raised concerns, which have substantially improved the manuscript.

In the manuscript entitled "The link of pancreatic iron with glucose metabolism and cardiac iron in thalassemia intermedia: a large, multicenter observational study" Meloni et al. aim to determine the link between pancreatic iron and glucidic metabolism, cardiac iron and complications through T2* MRI values. The study is well designed and provides novel insights regarding the use of pancreatic iron overload for clinical use in the context of thalassemia intermedia. The manuscript is generally well written, but this Reviewer has some minor remarks that should be considered before publication.

A: Thank you for this comment.

Since the aim of this study and its findings are quite novel, the abstract could be strengthened by the use of a small piece of “introductory” sentences (one or two sentences).

A: Thank you for this useful suggestion. We have added the following sentence in the Abstract: “In thalassemia major pancreatic iron was demonstrated a powerful predictor not only for the alterations of glucose metabolism but also for cardiac iron, fibrosis and complications, supporting a profound link between pancreatic iron and heart disease.”. Moreover, we have clearly stated that this is the first study evaluating  the prevalence of pancreatic iron overload in thalassemia intermedia and systematically exploring the link between pancreas T2* values and glucose metabolism and cardiac outcomes.

Although all abbreviations have been determined this Reviewer believes that in some sentences there is an extensive use of these abbreviations, making the manuscript more difficult to read. I believe that using a slightly smaller number of abbreviations would help the reader follow the manuscript.

A: We have reduced the number of abbreviations.

In Figure 3 please provide a title.

A: The title has been provided below the Figure.

Some AUCs are rather low to moderate (e.g., 0.71 and 0.61). In the second case, the authors have to be praised for admitting this low to moderate cutoff in the Discussion section. However, they have to point it out in the first case for the group of NT-TI patients as well.

A: We have now added in the Results the following sentences for NT-TI patients. “In the group of NT-TI patients, global pancreas T2* values were comparable be-tween patients with normal and altered glucose metabolism (29.85±11.41 ms vs 28.87±9.67 ms; P=0.755). The ROC curve did not reveal a global pancreas T2* threshold below which the probability of detecting an altered glucose metabolism increases significantly with satisfying sensitivity and specificity (AUC=0.52; P=0.823)”.

We have stated in the discussion that in RT-TI patients the AUC for the prediction of an altered glucose metabolism was rather low.

This Reviewer believes that some parentheses containing numbers in the Discussion section could be avoided to make the manuscript easier to follow.

A: We have eliminated some parentheses containing numbers.

Reviewer 2 Report

This is an interesting study that fills a gap in the phenotypic characterization of thalassemia intermedia. The studies are carefully performed, methodology is appropriate, as are the analysis of data and the conclusions drawn from those analyses.

There are some minor points which the authors need to address, mostly in matters of information presentation.

  1. On line 41, the authors should refer to “glucose metabolism” or “glycemic metabolism” rather than “glucidic”.

  1. The authors state that there are three primary phenotypes of beta thalassemia: major, intermediate, and minor. They describe major and intermedia. They should also insert a brief description of minor.

  1. In lines 91-92, there is something missing from the sentence. The authors say “… Glucose intolerance occurs in an earlier stage…” Earlier as compared to what?

  1. On line 354, the verb phrase “play a role” should be moved later in the sentence, perhaps following the concluding words “glucose uptake”, the insertion of a comma, and then “play a role”.

Author Response

We would like to thank the Editor and the Reviewer for their encouraging feedback and constructive critique and for the effort regarding this manuscript. We have addressed each of the raised concerns, which have substantially improved the manuscript.

Comments and Suggestions for Authors

This is an interesting study that fills a gap in the phenotypic characterization of thalassemia intermedia. The studies are carefully performed, methodology is appropriate, as are the analysis of data and the conclusions drawn from those analyses.

A: We thank the Reviewer for this comment

There are some minor points which the authors need to address, mostly in matters of information presentation.

On line 41, the authors should refer to “glucose metabolism” or “glycemic metabolism” rather than “glucidic”.

A:  We have now used “glucose metabolism” instead of “glucidic metabolism”.

The authors state that there are three primary phenotypes of beta thalassemia: major, intermediate, and minor. They describe major and intermedia. They should also insert a brief description of minor.

 A: Thank you for this suggestion. We have now added a brief description of thalassemia minor as follows. “Beta-thalassemia minor, also called carrier or trait, is associated with the inheritance of a single gene defect and it is usually asymptomatic with mild to moderate microcytic anemia”.

In lines 91-92, there is something missing from the sentence. The authors say “… Glucose intolerance occurs in an earlier stage…” Earlier as compared to what?

A: The sentence has been modified as follows: “Glucose intolerance occurs early during adolescence…”

On line 354, the verb phrase “play a role” should be moved later in the sentence, perhaps following the concluding words “glucose uptake”, the insertion of a comma, and then “play a role”.

A: The sentence has been modified as suggested.